# Rapid Reduction of Herbicide Susceptibility in Junglerice by Recurrent Selection with Sublethal Dose of Herbicides and Heat Stress

**Lariza Benedetti** [1], **Gulab Rangani** [2], **Vívian Ebeling Viana** [1], **Pâmela Carvalho-Moore** [2], **Aldo Merotto Jr.** [3], **Edinalvo Rabaioli Camargo** [1], **Luis Antonio de Avila** [1,*] and **Nilda Roma-Burgos** [2,*]

1   Crop Protection Graduate Program (Programa de Pós-Graduação em Fitossanidade), Federal University of Pelotas (Universidade Federal de Pelotas), Pelotas 96160-000, RS, Brazil; larizabenedetti13@hotmail.com (L.B.); vih.viana@gmail.com (V.E.V.); edinalvo_camargo@yahoo.com.br (E.R.C.)
2   Department of Crop, Soil and Environmental Sciences, University of Arkansas, Fayetteville, AR 72704, USA; grangani@uark.edu (G.R.); pcarvalh@email.uark.edu (P.C.-M.)
3   Crop Science Department (Departamento de Plantas de Lavoura), Federal University of Rio Grande do Sul (Universidade Federal do Rio Grande do Sul), Porto Alegre 91540-000, RS, Brazil; aldo.merotto@ufrgs.br
*   Correspondence: laavilabr@gmail.com (L.A.d.A.); nburgos@uark.edu (N.R.-B.)

**Abstract:** Global climate change, specifically rising temperature, can alter the molecular physiology of weedy plants. These changes affect herbicide efficacy and weed management. This research aimed to investigate the combined effect of heat stress (HS) and sublethal doses of herbicides (four active ingredients) on adaptive gene expression and efficacy of herbicide on *Echinochloa colona* (L.) Link (junglerice). Three factors were evaluated; factor A was *E. colona* generation (G0-original population from susceptible standard; G1 and G2 were progenies of recurrent selection), factor B was herbicide treatment (florpyrauxifen-benzyl, glufosinate-ammonium, imazethapyr, quinclorac and nontreated check) and factor C was HS (30 and 45 °C). The herbicides were applied at 0.125× the recommended dose. Recurrent exposure to HS, combined with sublethal doses of herbicides, favors the selection of plants less susceptible to the herbicide. Upregulation of defense (antioxidant) genes (*APX*: *Ascorbate peroxidase*), herbicide detoxification genes (*CYP450 family*: *Cytochrome P450*), stress acclimation genes (*HSP*: *Heat shock protein*, *TPP*: *Trehalose phosphate phosphatase* and *TPS*: *Trehalose phosphate synthase*) and genes related to herbicide conjugation (*UGT*: *UDP Glucosyltransferase*) was significant. The positive regulation of these genes may promote increased tolerance of *E. colona* to these herbicides.

**Keywords:** *Echinochloa colona*; climate change; high temperature; low-dose herbicide; weed resistance evolution; susceptibility; transcriptome

## 1. Introduction

Accelerated world population growth means intensifying demand for food, both in quantity and quality [1]. Food security is compromised, directly or indirectly, by the increasing complexity of pest problems (including weeds), limitations in production inputs, variable availability of technology and infrastructure—all of which are impacted by climate change [2–6]. Worldwide, the genus *Echinochloa* (mainly, *E. colona*, *E. crus-galli* and *E. phyllopogon*) includes the most troublesome weeds in rice [7,8]. *E. colona* also infests many major crops including corn, soybean, sorghum, cotton and sugarcane [9]. *E. colona* can cause between 1.5% and 100% yield losses in rice mainly due its rapid

growth, high competitive ability, high seed production potential and wide ecological range [10,11]. Herbicides are still the most important tools for weed management because of their cost-effectiveness, rapid action, ease of use and residual efficacy in the case of soil-active compounds [12]. However, the intensive and extensive use of herbicides exerts high selection pressure on weed populations, selecting rare individuals with the ability to survive or escape the herbicide treatment, leading to a gradual loss of herbicide efficacy [12,13].

The International Survey for Herbicide Resistance database depicts 502 cases (species × site of action) of herbicide-resistant weeds globally as of July 2020 and counting [9]. Weeds have evolved resistance to 23 of the 26 known herbicide sites of action involving 167 different herbicides [9]. The first case of herbicide-resistant weed of any species in this genus was *E. colona*, reported in 1987 in Costa Rica, and this was to propanil, a photosystem II inhibitor [9]. Currently, 26 unique cases of herbicide-resistant *E. colona* have been reported globally [9]. The *Echinochloa* genus (specifically, *E. colona*, *E. crus-galli* and *E. phyllopogon*) has been ranked among the top four most problematic weeds with rapidly increasing cases of multiple resistance [12,14].

Recurrent selection with sublethal doses of herbicides leads to resistance evolution. This has been demonstrated in *Lolium rigidum* [15–17], *Raphanus raphanistrum* L. [18] and *Amaranthus palmeri* [19]. 'Low-dose selection' of weed populations occurs in the field all the time [20]. Low-dose selection in crop production fields arises from insufficient coverage of some individuals partially covered by other plants; variations in per-plant dose due to differences in weed size, weed density, field topography or soil type; drift rates to populations on field edges and other biological, environmental or physical factors.

Maximum herbicide efficacy is highly dependent on optimum environmental conditions [21]. The occurrence of high temperature events and increasing temperatures projected by the Intergovernmental Panel on Climate Change (IPCC), certainly has major consequences for global agriculture including pronounced effects on weed management [22,23]. Data are starting to accumulate showing reduced efficacy of certain herbicides under high temperature. This reduced efficacy was seen for cyhalofop on *E. colona* [24], glyphosate on *Conyza canadensis* and *Chenopodium album* [25], and glyphosate on *E. colona* [26]. A decline in sensitivity to ACCase inhibitors under high temperature was reported by Matzrafi et al. [27], which could be attributed to increased herbicide metabolism. The effect of genotype × environment and gene × environment interactions on non-target-site resistance (NTSR) mechanisms is apparent. If high temperature reduces the efficacy of certain foliar herbicides, then the effect of recurrent 'low-dose exposure' on weed resistance evolution will be accelerated under increasing temperature. Logically, there would be crosstalk between these stress responses in plants.

Understanding how herbicides and climate change shape the transgenerational memory of weed populations imprinted in their genomic profile is important as this could help discover novel tools and strategies to mitigate the evolution of weed resistance to herbicides. We hypothesized that: (a) treatment with sublethal doses of herbicide on *E. colona* under high temperature selects plants with greater adaptability to such conditions, resulting in reduced susceptibility to chemical control, with transgenerational effects; and (b) the application of low herbicide doses at high temperature promotes changes in adaptations to abiotic stresses, across generations (i.e., upregulation of protection genes), which consequently also reduce sensitivity to herbicides. Therefore, the objective of this research was to study the joint effect of high temperature and sublethal doses of herbicides on *E. colona* susceptibility to herbicides across generations.

## 2. Materials and Methods

### 2.1. Plant Material

This study was conducted using seeds of susceptible *E. colona* (referred to hereafter as G0), collected in 2011 from a field in Prairie County, Arkansas, USA. G0 was then exposed to three successive cycles of recurrent selection with sublethal doses of herbicides under optimal or high temperature to produce G1 and G2. The procedure is described in the following section.

### 2.2. General Procedure for Population Generation

The treatments tested in this study included optimal (30 °C) and high temperature (45 °C) and four herbicides recommended for *E. colona* control in rice crop (Table 1). Seeds (G0) were planted into 50-cell trays (four per cell) containing commercial potting soil (Sun Gro Horticulture Canada Ltd., Vancouver, Canada). Approximately one week later, seedlings (one per pot) were transplanted into square pots (7.6 cm wide, 10.2 cm tall) containing a 1:3 mixture by volume of commercial potting soil and field soil (Captina silt loam-fine-silty, siliceous, active and mesic typic fragiudults). The experiment was performed in a complete randomized block design, with six replications in each cycle. The experimental unit was one pot containing one plant. The experiment with populations G1 and G2 followed the same methodology described for G0 (Figure 1).

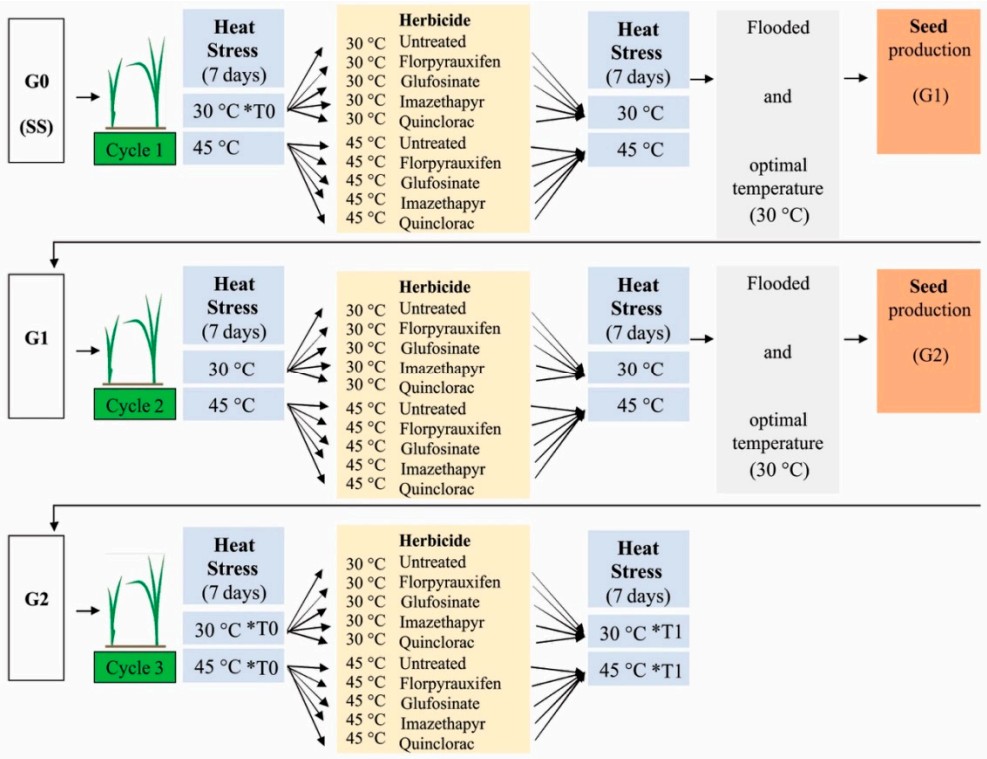

**Figure 1.** Schematic diagram of the progression of experiments on *Echinochloa colona* submitted to selection by low herbicides rates and heat stress starting with the parental population (G0) and selected (G1 and G2) progenies. Herbicide treatments were: nontreated (no herbicide), florpyrauxifen-benzyl, glufosinate-ammonium, imazethapyr and quinclorac, all applied at 0.125× the recommended dose. T0 (prior to herbicide application) and T1 (12 h after herbicide application) represent the timings of leaf tissue collection for RNA extraction and subsequent gene expression.

Preliminary experiments were conducted to determine the sublethal dose that would allow plants to survive and produce seeds. The experiment was conducted in the greenhouse at the University of Arkansas, Fayetteville, AR, USA with a 14-h photoperiod under a day/night temperature regime of 30 °C/21 °C. The pots were placed in trays and subirrigated until the seedlings reached the 2- to 3-leaf stage for treatment. The plants were then submitted to optimal temperature (30 °C) in the greenhouse and high temperature (45 °C) in the growth chamber for 7 d when the plants were sprayed with the respective herbicide treatments (Table 1). The heat-stressed plants were returned to (45 °C) for seven more days, after which all plants were grown under optimal temperature. The herbicides were applied in a spray chamber equipped with a flat fan spray nozzle (TeeJet spray nozzles; Spraying Systems Co., Wheaton, IL, USA), at a pressure of 221 kPa and a volume of 187 L ha$^{-1}$. After herbicide application, the plants were returned to the greenhouse or to the growth chamber. Watering by subirrigation was

resumed after 24 h. Three weeks after herbicide application, the herbicide effect was evaluated visually on a scale of 0% (no symptoms) to 100% (dead).

**Table 1.** Herbicide treatments used in the recurrent selection experiment with *Echinochloa colona* under concurrent high temperature stress.

| Active Ingredient | Trade Name | WSSA Group | MOA [a] | Chemical Family | Labeled Rate (g ai ha$^{-1}$) | Application Rate [b] (g ai ha$^{-1}$) |
|---|---|---|---|---|---|---|
| Florpyrauxifen-benzyl | Loyant™ | 4 | Synthetic auxins | Arylpicolinate | 30 | 3.75 |
| Glufosinate-ammonium | Liberty™ 280 SL | 10 | Inhibition of glutamine synthetase | Phosphinic acid | 448 | 56 |
| Imazethapyr | Newpath™ | 2 | Inhibition of acetolactate synthase | Imidazolinone | 211 | 26.38 |
| Quinclorac | Facet L™ | 4 | Synthetic auxins | Quinoline carboxylic acid | 432 | 54 |

[a] MOA: Mechanism of action and [b] application rate: all corresponding to 0.125× the recommended dose. Doses were based on preliminary experiments to allow survival and seed production (data not shown).

Prior to flowering, the plants were separated spatially to prevent cross-pollination. At maturity, seeds were harvested and bulked for each treatment. During each selection cycle, a separate group of six plants, without herbicide treatment, was grown to produce generations of plants exposed only to either optimal or high temperature. This procedure was repeated over three cycles.

*2.3. Determination of Susceptibility Level to Herbicides*

The original population (G0) and selected progenies (G1 and G2) were subjected to their respective heat treatment (30 or 45 °C) assignment, as implemented in the previous experiments, for 7 d and sprayed with a range of herbicide doses when the plants were at the 2- to 3-leaf stage, with two plants per treatment. The herbicide rates were 0×, 0.0625×, 0.125×, 0.25×, 0.5×, 0.75×, 1.0×, 1.5×, 2.0× and 4.0× the recommended dose. The herbicide treatments were applied as described previously. After herbicide application, the plants were returned to the greenhouse or growth chamber and were subirrigated again 24 h later. At three weeks after herbicide application, the herbicide effect was evaluated visually on a scale of 0% (no symptoms) to 100% (dead). Once the homoscedasticity and the normal distribution were verified, the data were subjected to analysis of variance (ANOVA). Dose response data were analyzed using the 'drc' package in R v. 3.3.0 (CRC Press Taylor & Francis Group: Boca Raton, FL, USA) [28,29]. The three-parameter log-logistic model in Equation (1) was used:

$$Y = d/1 + \exp[b(\log x - \log e)] \tag{1}$$

where $Y$ is the response (% control); $d$ is asymptotic value of $Y$ at the upper limit; $b$ is the slope of the curve around $e$ ($ED_{50}$: the herbicide rate giving response halfway between d and the lower asymptotic limit, which was set to 0) and $x$ is the herbicide rate. Susceptibility index (SI) was calculated as the ratio of the $ED_{50}$ of the G1 and G2 divided by the $ED_{50}$ of the G0 at 30 °C.

*2.4. Differential Gene Expression*

2.4.1. Plant Material

The control sample was composed of leaf tissues of G0 plants without herbicide treatment, grown under optimal temperature (30 °C), cultivated for three cycles, at the same period as the other plants to produce G2. In other words, the control samples were G0 plants that were cultured under optimal temperature without herbicide treatment. For all treatments, leaf tissues were collected before herbicide application (T0) and 12 h after herbicide application (T1). The tissues were flash-frozen in liquid nitrogen and stored in −80 °C until processed.

### 2.4.2. RNA Extraction and cDNA Synthesis

The RNA extraction process was performed using the reagent PureLinK[TM] (Plant RNA Reagent-Invitrogen[TM], Carlsbad, CA, USA), following the manufacturer's recommendations. The quantity and quality were assessed by agarose gel electrophoresis (1% *w/v*). The amount and purity of RNA were determined using a NanoDrop™ 2000 spectrophotometer (Thermo Scientific, Waltham, MA, USA). The RNA was treated with DNase and cDNA was synthesized using the RevertAid First Strand cDNA Synthesis Kit (Thermo Scientific, Waltham, MA, USA) according to the manufacturer's instructions.

### 2.4.3. qRT-PCR Assay

The qRT-PCR was conducted to determine changes in expression of NTSR candidate genes relative to the reference genes. The qRT-PCR was performed following MIQE guidelines [30], using specific oligonucleotides for target genes and the reference genes *ACT1*, *UBQ5* and *EF1-α* (Table 2). The qRT-PCR experiments were conducted in a total volume of 10 μL containing 5.0 μL of iTaq™ Universal SYBR[®] Green Supermix (Bio-Rad, Hercules, CA, USA), 0.5 μL primer forward (10 mM), 0.5 μL primer reverse (10 mM), 1 μL cDNA (1:5 dilution) and 3.0 μL of water. Two biological replicates and two technical replicates were performed for each treatment and primer pair.

**Table 2.** Oligonucleotides used for gene expression assay by qRT-PCR of *Echinochloa colona* that have undergone three cycles of recurrent selection with sublethal doses of herbicides under concurrent high temperature stress.

| Gene | Family/Name | Oligonucleotide Sequences (5′→3′) | Reference |
|---|---|---|---|
| *Act1* (Reference gene) | *Actin 1* | F: ATCCTTGTATGCTAGCGGTCGA<br>R: ATCCAACCGGAGGATAGCATG | [31] |
| *EF1-α* (Reference gene) | *Elongation factor 1α* | F: GTCATTGGCCACGTCGACTC<br>R: TGTTCATCTCAGCGGCTTCC | [31] |
| *UBQ5* (Reference gene) | *Ubiquitin 5* | F: ACCACTTCGACCGCCACTACT<br>R: ACGCCTAAGCCTGCTGGTT | [32] |
| *APX2* | *Ascorbate peroxidase* | F: CATCCTCTCCTACGCCGAC<br>R: CCTTCAGGAGGAGGCTCAG | [33] |
| *CYP709B1* | *Cytochrome P450 709B1* | F: GTCGTCAAGCAGGTGCTCTT<br>R: CAGTGAGGACGAGACCCTTG | [34] |
| *CYP709B2* | *Cytochrome P450 709B2* | F: GCCTGAGAGGTTCGAGTACG<br>R: CGATCATCGCAAAGTTCTGA | [34] |
| *CYP72A14* | *Cytochrome P450 72A14* | F: TCGGTGGCATCAAATATCCT<br>R: GAACTTGCCTGCGTCTTTTC | [34] |
| *CYP72A15* | *Cytochrome P450 72A15* | F: CCAGTGAGCTGATACGCAGA<br>R: GACGTCGCCTGTGAGATTTT | [34] |
| *UGT75D* | *Glycosyltransferase* | F: GCTCACTTTCCCGTTCCAG<br>R: GTGGTGGAGAATGTGACGAG | [34] |
| *HSP10* | *Heat shock protein 71.10* | F: CCGTGTGCTTCGACATTGAC<br>R: CGTTGGTGATGGTGHTCTTGTT | [35] |
| *HSP15* | *Heat shock protein 24.15* | F: GATCAAGGCGGAGATGAAGAAC<br>R: ACTCGACGTTGACCTGGAAGA | [35] |
| *TPP* | *Trehalose phosphate phosphatase* | F: TTGAAGGTGCGAGTGTTGAG<br>R: AACCACTCCCCAGTCCTTCT | [34] |
| *TPS* | *Trehalose phosphate synthase* | F: ACAGAGGGGCTACATTGCAC<br>R: CTGCAACTGCTCCAAGTGAA | [34] |

The amplification efficiency was determined for each primer pair and melt-curve analysis were performed. The cycle conditions were amplified for one cycle of 95 °C for 5 min, followed by 39 cycles of denaturation at 95 °C for 15 s, annealing at 60 °C for 1 min and extension at 72 °C for 30 s and a final dissociation curve of denaturation at 95 °C for 5 s, followed by cooling to 70 °C for 1 min and gradual heating at 0.11 °C steps to 95 °C and cooling 40 °C for 30 s. For each gene analyzed, *UBQ5* was used as an endogenous control to quantify cDNA abundance, after the stability analysis of the expression data using DataAssist™ v3.0 Software (Applied biosystems, Life Technologies, Carlsbad, CA, USA).

Cycle threshold (Ct) during the reaction cycles, and relative gene expression values were calculated using the $2^{-\Delta\Delta Ct}$ method [36]. Gene expression data were analyzed using MultiExperiment Viewer (MeV) software (The Institute for Genomic Research (TIGR), J. Craig Venter Institute, USA) and presented as a heat map diagram using the harvest stage as a baseline [37]. mRNA abundance of each gene from the G0 population (control from susceptible standard) served as the baseline for determining relative RNA levels, in the G2 population treatment.

## 3. Results

### 3.1. Determination of Susceptibility Level to Herbicides

Dose response assays were performed to determine the level of sensitivity after the herbicide-susceptible *Echinochloa colona* parental population was treated with sublethal doses of florpyrauxifen-benzyl, glufosinate-ammonium, imazethapyr or quinclorac under optimal or high temperature, over three consecutive cycles. Recurrent exposure to high temperature stress, combined with low-dose herbicide stress, reduced the sensitivity of *E. colona* to the herbicides tested (Table 3). Under optimal temperature (30 °C) the $ED_{50}$ increased minimally in general between G0 and G2 with all herbicides tested. Specifically, the $ED_{50}$ increased by 0.12 with florpyrauxifen-benzyl (Figure 2A–C); 5.28 with glufosinate-ammonium (Figure 2D–F); 0.8 with imazethapyr (Figure 2G–I) and 0.5 with quinclorac (Figure 2J–L). Thus, the susceptibility index remained unchanged (Table 3).

**Table 3.** Parameter estimates (*b*, *d*, $ED_{50}$ and SI) for the control of *Echinochloa colona* with various herbicides across cycles of exposure to high temperature stress and sublethal herbicide dose, three weeks after herbicide treatment. The data were fitted with a three-parameter log-logistic regression model in Equation (1).

| Log-Logistic Regression Estimates [a] | | | | | |
|---|---|---|---|---|---|
| Treatments [b] | *b* | *d* | $ED_{50}$ | *p* Value [c] | SI [d] |
| **Florpyrauxifen-benzyl** | | | (g ae ha$^{-1}$) | | |
| G0·30 °C | −1.99 (0.08) | 100.79 (0.67) | 3.34 (0.07) | | - |
| G0·45 °C | −1.36 (0.06) | 103.93 (0.95) | 3.91 (0.12) | 0.00053 | 1.17 |
| G1·30 °C | −2.23 (0.10) | 100.28 (0.61) | 3.46 (0.07) | | 1.04 |
| G1·45 °C | −1.69 (0.07) | 101.34 (0.81) | 4.52 (0.11) | $2.6685 \times 10^{-11}$ | 1.35 |
| G2·30 °C | −2.23 (0.08) | 100.28 (0.49) | 3.46 (0.06) | | 1.04 |
| G2·45 °C | −1.81 (0.06) | 102.08 (0.66) | 5.69 (0.11) | $2.5105 \times 10^{-24}$ | 1.70 |
| **Glufosinate-ammonium** | | | (g ae ha$^{-1}$) | | |
| G0·30 °C | −1.67 (0.09) | 101.89 (0.98) | 53.65 (1.73) | | - |
| G0·45 °C | −1.47 (0.08) | 102.65 (1.13) | 56.95 (2.07) | 0.214212 | 1.06 |
| G1·30 °C | −1.85 (0.11) | 101.58 (1.11) | 58.93 (2.11) | | 1.10 |
| G1·45 °C | −1.43 (0.08) | 104.00 (1.58) | 81.00 (3.76) | $2.7443 \times 10^{-7}$ | 1.51 |
| G2·30 °C | −1.85 (0.16) | 101.58 (1.52) | 58.93 (2.89) | | 1.10 |
| G2·45 °C | −1.30 (0.10) | 106.35 (2.72) | 109.31 (7.93) | $2.1293 \times 10^{-11}$ | 2.04 |
| **Imazethapyr** | | | (g ai ha$^{-1}$) | | |
| G0·30 °C | −2.83 (0.32) | 98.20 (1.31) | 38.60 (1.37) | | - |
| G0·45 °C | −1.83 (0.14) | 102.23 (1.60) | 39.98 (1.83) | 0.54070 | 1.04 |
| G1·30 °C | −3.46 (0.49) | 97.01 (1.40) | 39.53 (1.44) | | 1.02 |
| G1·45 °C | −1.95 (0.16) | 102.55 (1.95) | 51.98 (2.65) | $1.6142 \times 10^{-5}$ | 1.35 |
| G2·30 °C | −3.47 (0.54) | 96.57 (1.49) | 39.40 (1.54) | | 1.02 |
| G2·45 °C | −2.03 (0.17) | 102.61 (2.17) | 61.18 (3.29) | $1.0319 \times 10^{-9}$ | 1.58 |

**Table 3.** *Cont.*

| Treatments [b] | b | d | ED$_{50}$ | p Value [c] | SI [d] |
|---|---|---|---|---|---|
| Quinclorac | | | (g ae ha$^{-1}$) | | |
| G0·30 °C | −1.62 (0.13) | 104.90 (2.25) | 100.44 (6.07) | | - |
| G0·45 °C | −1.86 (0.16) | 104.69 (2.12) | 116.99 (6.24) | 0.04857 | 1.16 |
| G1·30 °C | −1.60 (0.12) | 104.88 (2.01) | 100.94 (5.44) | | 1.01 |
| G1·45 °C | −2.01 (0.16) | 104.20 (1.89) | 133.34 (5.93) | $5.7162 \times 10^{-5}$ | 1.33 |
| G2·30 °C | −1.60 (0.10) | 104.88 (1.69) | 100.94 (4.56) | | 1.01 |
| G2·45 °C | −2.24 (0.15) | 102.71 (1.54) | 139.53 (4.95) | $1.2312 \times 10^{-7}$ | 1.39 |

[a] Values in parenthesis are standard errors of the mean. [b] The temperature stress treatments were 30 and 45 °C; G0 indicates the first cycle of treatments, G1 and G2 indicate the subsequent generations. [c] p value: comparing the difference between optimal and high temperature at the same cycle by the SI function in the 'drc' package in R v. 3.3.0. [d] SI: susceptibility index was calculated as the ratio of the ED$_{50}$ divided by the ED$_{50}$ of the parental population (G0) at 30 °C.

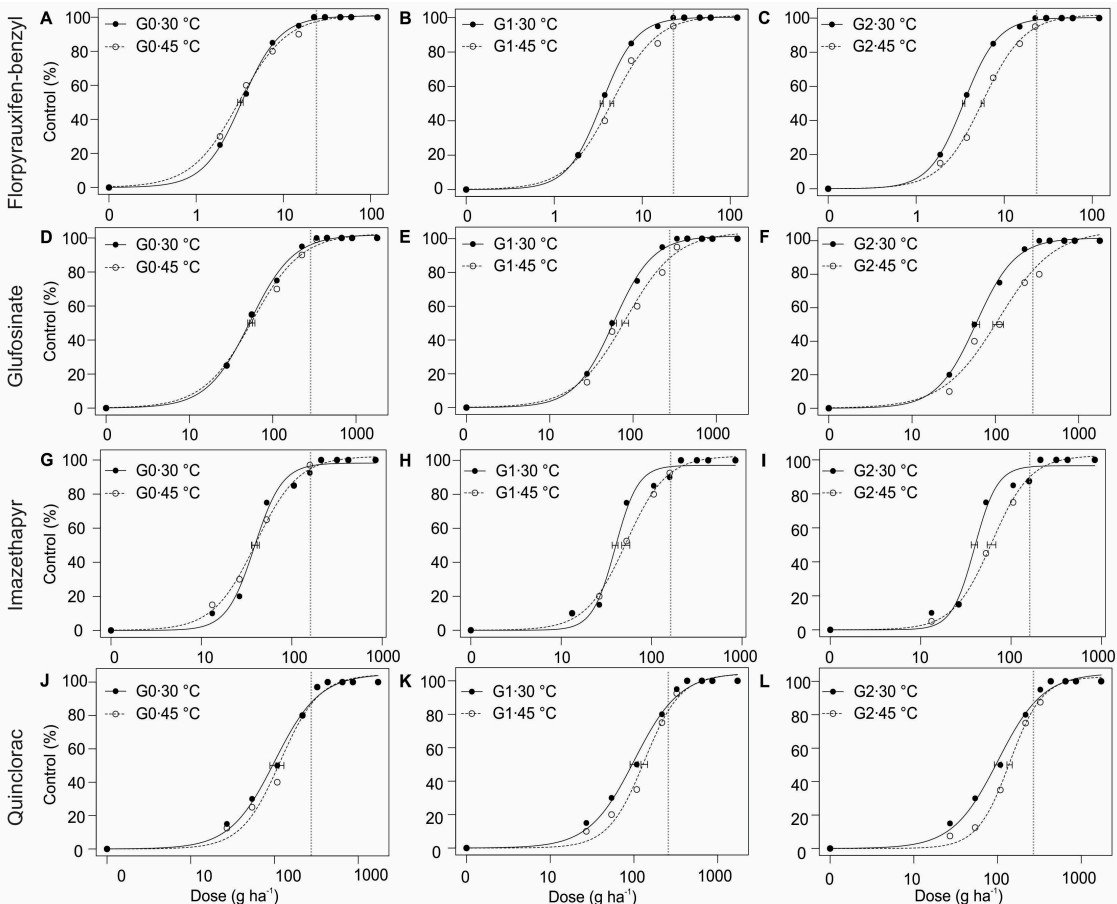

**Figure 2.** Nonlinear regression analysis of herbicide efficacy on *Echinochloa colona* across cycles of recurrent exposure to high temperature stress and the sublethal herbicide dose. The herbicide treatments were: (**A–C**) florpyrauxifen-benzyl; (**D–F**) glufosinate-ammonium; (**G–I**) imazethapyr and (**J–L**) quinclorac. G0 = original population; G1 and G2 are progenies after progressive cycles of selection. Weed control was evaluated three weeks after herbicide application. Each data point is an average of two replicates. The data were fitted with a three-parameter log-logistic model in Equation (1). Vertical dotted gray lines indicate the recommended dose of each herbicide.

Under high-temperature stress (45 °C), the ED$_{50}$ increased significantly between G0 and G2 with all herbicides (Table 3). These increases were higher compared to the ED$_{50}$ increases under optimal temperature across generations. The ED$_{50}$ increased 21.68 with florpyrauxifen-benzyl (Figure 2A–C);

52.4 with glufosinate-ammonium (Figure 2D–F); 21.2 with imazethapyr (Figure 2G–I) and 18.4 with quinclorac (Figure 2J–L). The increase in tolerance was 10–37 times higher under high temperature stress compared to optimal temperature.

*3.2. Differential Gene Expression*

To understand the abiotic stress tolerance response between generations subjected to high temperature stress and herbicides, genes related to oxidative stress (*APX*), herbicide detoxification (*CYP450 family*), heat stress (*HSPs family*), abiotic stress response and tolerance (*TPP* and *TPS*) and herbicide molecule conjugation (*UGT*) were analyzed (Figure 3). *E. colona* plants (G2) exposed to high-temperature stress over three cycles showed increased expression of *UGT* (0.7-fold), *TPP* (0.9-fold), *TPS* (1.6-fold), *CYP72A14* (2.0-fold), *CYP709B1* (5.1-fold) and with the highest relative expression observed in *CYP709B2* gene (6.4-fold), compared to G0 plants at optimal temperature (Figure 3).

3.2.1. Florpyrauxifen-Benzyl

The relative expression of some target genes increased in G2 compared to G0, after recurrent exposure to a low dose of florpyrauxifen-benzyl, with or without heat stress. The *APX* gene was constitutively expressed but was upregulated in G2 12 h after florpyrauxifen-benzyl application (Figure 3). The expression of *CYP72A15* in G2 plants selected with florpyrauxifen-benzyl was not different between plants kept at 30 °C and those in 45 °C for seven days, without florpyrauxifen-benzyl treatment. Upon treatment with florpyrauxifen-benzyl, *CYP72A15* was induced 4.5-fold in G2 plants kept at high temperature and downregulated (−0.1-fold) in optimal temperature condition. Florpyrauxifen-benzyl induced the expression of *CYP72A14*, but the induction was greater under optimal temperature (4.2-fold) than at high temperature (3.2-fold). The other two genes of the cytochrome P450 family showed a similar profile (Figure 3) without florpyrauxifen-benzyl treatment. *CYP709B1* and *CYP709B2* were not differentially expressed under 30 °C, but both genes were highly induced at 45 °C (6.1- and 8.4-fold, respectively). Both genes were downregulated after treatment with florpyrauxifen-benzyl (4.9- and 2.0-fold, respectively).

The *HSP10* and *HSP15* genes are members of the heat shock protein family. The greatest induction of *HSP10* occurred at 30 °C with values of 0.9-fold and 3.1-fold at T0 and T1, respectively (Figure 3). *HSP10* was not induced under heat stress, either before or after treatment with florpyrauxifen-benzyl. Similar to *HSP10*, *HSP15* was induced 12 h after florpyrauxifen-benzyl application with values of 7.7-fold and 3.8-fold at 30 °C and 45 °C, respectively.

Concerning the TPS gene, induction of 1.6-fold was detected when plants were heat-stressed, and plants which received heat and florpyrauxifen-benzyl treatment (12 h) a 0.2-fold expression was detected. These findings indicated the possible involvement of the *TPS* gene with abiotic stress tolerance. For the *TPP* gene the highest value was observed at 30 °C after the application of florpyrauxifen-benzyl (2.7-fold); however, at 45 °C before the application it was 1.6 and 0.8 in T1. The *UGT* gene, which is related to herbicide conjugation, increased its expression from −1.4 in T0 to 3.9 in T1 at 30 °C and from 2.3 in T0 to 3.7 in T1 at 45 °C (Figure 3).

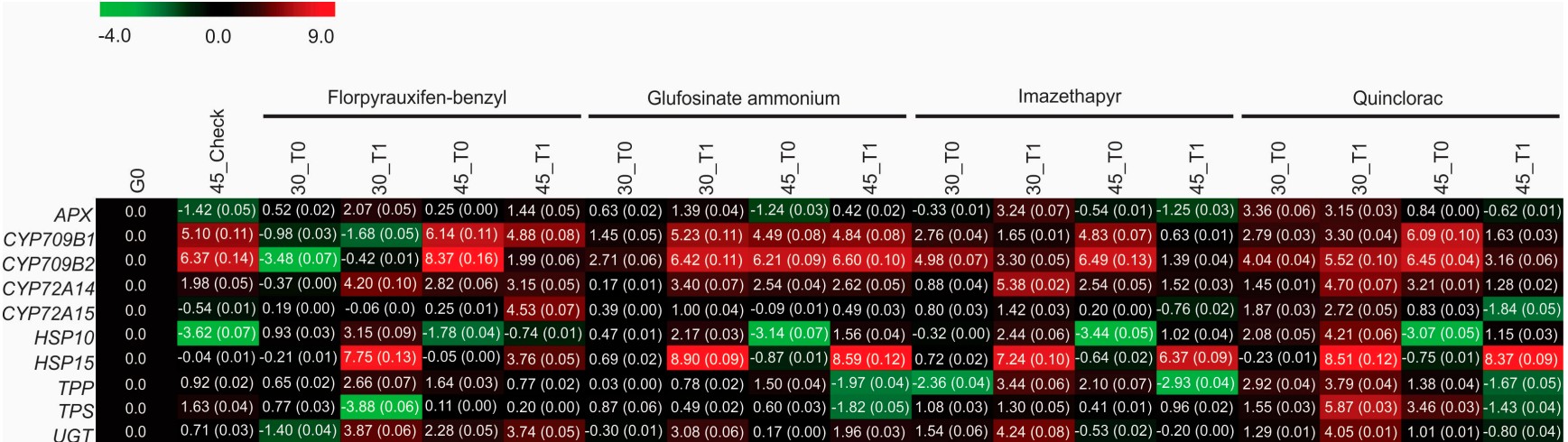

**Figure 3.** Relative mRNA abundance in Junglerice (*Echinochloa colona*) leaves. mRNA abundance is represented in Log2 fold-change using the Multi Experiment Viewer (TIGR MeV) software [37] and values in parentheses represent 95% confidence interval. mRNA abundance of each gene from G0 population (original susceptible standard) served as the baseline for determining relative RNA levels. The numbers 30 and 45 are the temperature treatments in °C; untreated = G2 without herbicide; T0 = before herbicide application; T1 = 12 h after herbicide application to G2 plants. The color scale below the heatmap shows the expression level; red = high transcript abundance, green = low transcript abundance.

### 3.2.2. Glufosinate-Ammonium

The *APX* gene was induced 12 h after glufosinate-ammonium application with values ranging from 0.6- to 1.4-fold and −1.2- to 0.4-fold at 30 and 45 °C, respectively (Figure 3). With *HSP* genes, specifically *HSP15* the expression level did not change (8.9-fold at 30 °C and 8.6-fold at 45 °C), after glufosinate-ammonium treatment.

The *TPS* and *TPP* genes were significantly downregulated 12 h after glufosinate-ammonium treatment under 45 °C, while *UGT* was upregulated under the same treatment (Figure 3). This indicates possible use of the substrate from the trehalose biosynthesis pathway for conjugation with the herbicide. The expression of *CYP450* genes did not change between G0 and G2, nor was it affected by glufosinate-ammonium treatment under optimal temperature conditions. However, the *CYP450* genes *CYP709B1*, *CYP709B2*, *CYP72A14* and *CYP72A15* were upregulated under high temperature 12 h after glufosinate-ammonium treatment, also indicating a possible attempt to detoxify the herbicide.

### 3.2.3. Imazethapyr

Imazethapyr induced the expression of *APX* (3.2-fold) 12 h after treatment under optimal temperature, but suppressed *APX* under high temperature (−1.2-fold; Figure 3). The heat shock protein genes, *HSP10* and *HSP15*, were induced by imazethapyr, under optimal and high temperatures, specifically *HSP15* with expression greater than 7.2-fold at 30 °C and 6.4-fold at 45 °C.

Under optimal temperature, *TPS* (1.1- to 1.3-fold), TPP (−2.4- to 3.4-fold) and *UGT* (1.5- to 4.2-fold) were upregulated 12 h after imazethapyr application. Additionally, *TPS* (0.4- to 1.0-fold) was upregulated at 45 °C, but to a lesser extent (Figure 3). *TPP* and *UGT* were downregulated at 45 °C 12 h after imazethapyr treatment, with an expression of −2.9- and −0.2-fold, respectively.

*CYP709B1* and *CYP709B2* were highly expressed at 45 °C in G2 plants before herbicide (4.8- and 6.5-fold, respectively), but expression was significantly reduced 12 h after imazethapyr application (0.6- and 1.4-fold; Figure 3). The same gene expression pattern was observed at optimal temperature with *CYP709B1* expression being downregulated from 2.8- to 1.7-fold and *CYP709B2* expression being downregulated from 5.0- to 3.3-fold after imazethapyr treatment. However, *CYP72A14* and *CYP72A15* were induced 12 h after imazethapyr treatment at 30 °C, indicating the possible involvement of these two genes with imazethapyr detoxification under optimal temperature conditions. As observed with *CYP709B1* and *CYP709B2*, the *CYP72A14* and *CYP72A15* genes were also downregulated after imazethapyr treatment at high temperature.

### 3.2.4. Quinclorac

The *APX* antioxidant enzyme-encoding gene expression did not change 12 h after quinclorac treatment under optimal temperature, but was downregulated (from 0.8- to −0.6-fold) under high temperature (Figure 3). On the other hand, *HSP10* and *HSP15* were upregulated after quinclorac treatment mainly under optimal temperature. The combination of quinclorac plus high temperature also leads to an upregulation of *HSP* genes; however, lower transcript accumulation was detected compared to optimal temperature. These results indicate that heat shock proteins are most likely to play a role in mitigating the effect of quinclorac.

Under optimal temperature, *TPS*, *TPP* and *UGT* genes were induced 12 h after quinclorac application, showing 5.9-, 3.8- and 4.1-fold expressions, respectively (Figure 3). This indicated the possible role of these genes in coping with quinclorac stress. However, these genes were downregulated after the plants were exposed to quinclorac and heat stress treatment, showing −1.4-, −1.7- and −0.8-fold expression, respectively. Similarly, *CYP450*-encoding genes were induced after quinclorac application under optimal temperature and downregulated at 45 °C.

In general, this study highlighted some candidate genes related to herbicide detoxification and/or stress tolerance mechanisms that help promote the adaptive evolution of plants such as *E. colona* to heat stress and herbicides. Some candidate genes are; *APX*, *CYP72A14*, *CYP72A15*, *HSP15*, *TPS* and

*UGT* for florpyrauxifen-benzyl-benzyl; *APX*, *CYP709B2*, *CYP72A14*, *CYP72A15*, *HSP10*, *HSP15* and *UGT* for glufosinate-ammonium; *HSP10*, *HSP15*, *TPS* and *UGT* for imazethapyr and *HSP10* and *HSP15* for quinclorac.

## 4. Discussion

### 4.1. Heat Stress Reduces Susceptibility of Junglerice to Herbicides

Temperature governs plant growth and development, herbicide uptake and translocation and herbicide detoxification and protection processes thereby influencing herbicide efficacy and evolution of weed resistance to herbicides [26,38–40]. This research supports our hypothesis that recurrent selection of *E. colona* with a sublethal dose of herbicides under high temperature increases plant tolerance to the herbicide, with transgenerational effects. The increase in the level of tolerance to the herbicides tested was still below the recommended dose (Table 3), but the accelerated evolution of higher tolerance under high temperature was apparent. The 'resistance evolution rate' observed here was not higher than was reported for either *Lolium rigidum* with low doses of the herbicide diclofop-methyl for two generations or *Raphanus raphanistrum* following four generations of low doses of 2,4-D selection [16,18]. It is important to highlight that *E. colona* is a self-pollinating species and evolution may be slower due to the low crossing rate, unlike allogamous species where genetic traits of resistance can accumulate rapidly during crossing and in successive generations from surviving plants [15,41,42]. Moreover, previous studies also suggested that abiotic stresses, such as herbicides (propanil, quinclorac, imazethapyr and cyhalofop), elevated temperature (26/38 °C night/day) and elevated $CO_2$ concentration (700 ± 50 mmol·mol$^{-1}$) favor the selection of *Echinochloa* spp. plants resistant to herbicides [14,24], corroborating the results reported in this study.

### 4.2. Gene Expression Profile Reflects the Transgenerational Reduces Susceptibility Memory

In the plant-environment system there is a dynamic signaling and response process resulting from the metabolic capacity that results in a series of molecular, physiological, biochemical and morphological changes [43–46]. At the molecular level, the complex signaling system reprograms gene expression, prioritizing stress-responsive genes so that response in plants can be characterized as increased tolerance to a subsequent stressful event [47–50]. In the literature there are suggestions that epigenetic regulation is also involved in the evolution of resistance, based on the response of oxidative stress arising from herbicides, as is seen with abiotic stresses in plants [51–53].

The ability to upregulate genes to detoxify a range of chemicals in a plant is a result of increased metabolism that is directly affecting increased tolerance and, eventually resistance, to herbicides. Such increased metabolism is strongly influenced by climate change, which can accelerate the adaptation process in both crops and weeds [54–58]. The complexity of the system and the data scarcity on weed species make it difficult to understand the herbicide metabolism mechanisms, as there is a great variability of candidate genes and processes that may be involved [59].

During the development of an organism, epigenetic mechanisms establish patterns of gene expression to ensure adequate differentiation, adaptation to environmental changes and future responsiveness due to previous exposure to particular conditions over the short and long term, even affecting subsequent generations [60,61]. In this context, the APX enzyme plays an essential role in the antioxidant defense system, acting to prevent oxidative damage caused by ROS (reactive oxygen species) [62,63]. The *APX* gene has been reported to respond to thermal stress, endowing protection from herbicide effects, as corroborated by the increased expression seen in the current study (Figure 3). This response was previously reported after florpyrauxifen-benzyl and glufosinate-ammonium application, indicating a potential mechanism for adaptation to abiotic stress related to the mechanisms of action (MOAs) of these active ingredients [62,64–66]. On the other hand, a reduction in *APX* expression observed 12 h after quinclorac application was also observed in the shoots of

quinclorac-sensitive *Echinochloa oryzicola* [40,67]. This reduction indicates that the adaptation of *Echinochloa colona* at low quinclorac use rates and high temperature is not a result of *APX* expression.

Herbicide detoxification is a complex metabolic process involving herbicide transformation, conjugation and exportation for final processing, coordinated by enzymes and regulatory genes [55,68]. Herbicide metabolism studies indicate that tolerance or resistance is mostly due to increased rates of herbicide detoxification, in both crops and weeds, and generally involves the activity of cytochrome P450 monooxygenases [12,14,69–73]. In this study, the expression of *CYP72A14* and *CYP72A15* increased when exposed to florpyrauxifen-benzyl-benzyl, while *CYP709B1*, *CYP709B2*, *CYP72A14* and *CYP72A15* were induced by glufosinate-ammonium treatment, especially when subjected to high temperature stress. Cytochrome P450s are phase 1 detoxification enzymes. The induction of the genes mentioned, by florpyrauxifen-benzyl and glufosinate, indicates some level of metabolic degradation of these herbicides in susceptible *E. colona*, although such a process would not occur fast enough to prevent plant death. However, higher induction of these genes after herbicide treatment under high temperature indicates that the degradation process could occur slightly faster, effecting some elevation in tolerance to the herbicide. Recurrent exposure to these conditions apparently has some transgenerational footprint; thus, we observed increased $ED_{50}$ values in G2 plants under high temperature relative to G0. Herbicide metabolism mediated by increased *CYP450* expression was reported in resistant plants such as *Alopecurus aequalis*, *Arabidopsis*, *Echinochloa phyllopogon*, *Glycine max*, *Lolium rigidum* and *Oryza sativa* [73–77].

The reduction of *CYP450* gene expression observed with imazethapyr and quinclorac may be due to a variety of reasons. First, the mechanisms by which *E. colona* copes with these herbicides do not involve these specific *CytP450* genes. The CYP450 family is large and diverse (350,000 cytochrome P450 sequences) [78,79]. It is possible that other *CytP450* genes are involved, which are not included in the selection of primary target genes. Dalazen et al. [80] demonstrated the importance of degradation enhancement for resistance of *Echinochloa crus-galli* to imazethapyr in terms of relatively high expression of CYP81A6 and GSTF1 genes. Wright et al. [59] studying a biotype of junglerice with resistance to four herbicides (imazamox, fenoxaprop-P-ethyl, quinclorac and propanil) demonstrated that for two of these herbicides (imazamox and quinclorac) the resistance was reduced when the herbicide was applied with a cytochrome P450 inhibitor (malathion). Therefore, the differential expression analysis and qPCR were performed, but did not identify any cytochrome P450s as differentially expressed; they did identify a *GST* and *kinase* as being significantly upregulated. Other coping mechanisms might be involved, as demonstrated by Chayapakdee et al. [81] where *Echinochloa phyllopogon* resistance to quinclorac was associated with reduced ethylene synthesis rather than enhanced cyanide detoxification by β-cyanoalanine. On the other hand, the *EcCAS* gene mutation and higher gene expression may enhance the activity of β-CAS to avoid the accumulation of toxic cyanide in resistant populations, thus contributing to the resistance mechanism of *E. crus-galli* var. *zelayensis* to quinclorac [82]. Thus, there are many possibilities involved in the pathways of tolerance and resistance of *Echinochloa* spp. to herbicides.

Plants are exposed to various biotic and abiotic conditions that, when exceeding the limits of fluctuations, can be deemed as stress [83,84]. The action of *HSPs* can be activated under various stress conditions including osmotic and oxidative stress, drought, salinity, extreme temperatures, pesticide response and multiple stress resistance, among others [25,85–89]. The upregulation of *HSP* 12 h after application of all herbicides under both temperature treatments (30 and 45 °C), especially *HSP15*, indicates possible contribution to the adaptation to herbicides with different MOAs and thermotolerance. HSPs are molecular chaperones particularly important for plant survival, playing a large role in many processes that regulate protein molding/folding, localization, accumulation and degradation, and assisting in cellular homeostasis under ideal and adverse conditions [35,90,91].

In response to several environmental stresses, another process important to plant adaptation is the trehalose biosynthetic pathway and trehalose accumulation. Here, the *TPP* gene was upregulated at 30 °C in response to all herbicides, but was downregulated after herbicide treatment at 45 °C.

This suggests the use of trehalose substrate for another purpose that does not need to flow through the action of TPP [34,91–93]. Another component of the trehalose pathway, TPS, was also induced under high temperature stress after application of florpyrauxifen-benzyl and imazethapyr. The trehalose pathway could have a significant role in the evolution of NTSR mechanisms to florpyrauxifen-benzyl and imazethapyr in *E. colona*. However, *TPS* expression declined after application of glufosinate-ammonium and quinclorac at 45 °C. Thus, *TPP* and *TPS* gene expression appears to be inhibited by high temperature and the application of glufosinate-ammonium and quinclorac, indicating that these genes are not involved in tolerance to thermal stress combined with these particular herbicides. These results corroborate previous reports about HSPs and trehalose being important in plant adaptation and tolerance to various environmental stresses [94–101].

*UDP-glucosyltransferases* (*UGT*) is another group of genes involved in xenobiotic detoxification, which is also associated with auxin synthesis, signaling and response [86]. *UGT* may play a role in plant response to florpyrauxifen-benzyl as it was induced by the herbicide under both temperatures (Figure 3). Increased *UGT* expression was also observed after glufosinate-ammonium and imazethapyr application. Enzymes in the glycosyltransferase family have been reported to play a role in phase II of xenobiotic detoxification, specifically herbicide metabolism, and are associated with abiotic stress tolerance [55,69,102,103].

The study of adaptation mechanisms to environmental and herbicide stress in *Echinochloa colona* is important since plants are constantly exposed to such stresses. Concurrent exposure to environmental stress can facilitate the evolution of tolerance to new herbicides and different MOAs, as demonstrated by Benedetti et al. [104] where observed that recurrent selection by a herbicide sublethal dose and drought stress results in rapid reduction of herbicide sensitivity in junglerice. This research provides information on the reduction of sensitivity to herbicides under high temperature and, thus, the imminent acceleration of weed resistance evolution to herbicides under heat stress (Figure 4). However, information is still needed on the genetic, biochemical or physiological mechanisms that endow resistance. Moreover, it would be valuable to study the combination of different stress factors such as herbicides and temperatures to understand responses to multiple stress factors.

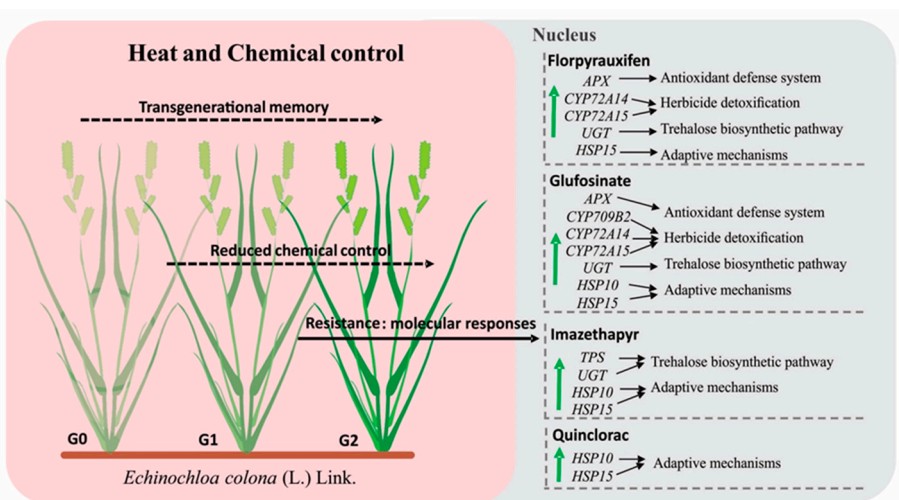

**Figure 4.** Summary of the response of *Echinochloa colona* to recurrent selection by sublethal doses of herbicides and heat stress.

The increased expression of genes related to stress signaling and herbicide metabolism, appear to be involved in response to mediate the negative effects by herbicide activity and heat stress, and can be a facilitator of adaptation of *E. colona* (Figure 4). Research on *E. colona* populations that already evolved resistance to these herbicides would be useful in providing further understanding of adaptation mechanisms to environmental stresses.

## 5. Conclusions

The joint effect of recurrent selection with a sublethal dose of herbicide and heat stress reduced junglerice susceptibility, due to adaptive expression of metabolism- and protection-related genes in *E. colona* with a transgenerational effect.

The significant upregulation of defense (antioxidant) genes (*APX*: *Ascorbate peroxidase*), herbicide detoxification genes (*CYP450 family*: *Cytochrome P450*), stress acclimation genes (*HSP*: *Heat shock protein*, *TPP*: *Trehalose phosphate phosphatase* and *TPS*: *Trehalose phosphate synthase*) and genes related to herbicide conjugation (*UGT*: *UDP Glucosyltransferase*) may promote reduction in sensitivity of *E. colona* to herbicides under climate change.

**Author Contributions:** Conceptualization, L.B., N.R.-B., L.A.d.A., and G.R.; data curation, L.B., G.R., and V.E.V.; formal analysis, L.B. and V.E.V.; funding acquisition, L.A.d.A. and N.R.-B.; investigation, L.B., G.R., P.C.-M., and N.R.-B.; methodology, L.B., G.R., N.R.-B., V.E.V., P.C.-M., A.M.J., E.R.C., and L.A.d.A.; project administration, N.R.-B. and L.A.d.A.; resources, N.R.-B., and L.A.d.A.; Supervision, A.M.J., E.R.C., L.A.d.A., and N.R.-B.; validation, L.B. and G.R.; visualization, L.B. and V.E.V.; writing—original draft, L.B.; writing—review and editing, N.R.-B., L.B., G.R., V.E.V., P.C.-M., A.M.J., E.R.C., and L.A.d.A. All authors have read and agreed to the published version of the manuscript.

**Funding:** This research received funding from: Coordenação de Aperfeiçoamento de Pessoal de Nível Superior-Brasil (CAPES)-Finance Code 001; BASF Corporation grant to N.R.-B.; the University of Arkansas (Hatch Project ARK02416), Fayetteville, NC, USA; the research fellowship of L.A.d.A. by Conselho Nacional de Desenvolvimento Científico e Tecnológico (CNPq—Proc.N. 310538/2015–7) and the student sandwich doctoral fellowship of L.B. was financed by the CNPq—Proc.N. 208443/2017–7 CNPq. This research was made possible by the direct financing from the "Ciência Sem Fronteiras" public call MEC/MCTI/CAPES/CNPQ/FAPS-Visiting Researcher fellowship-PVE 2014 (Public Call number 401381/2014–5).

**Acknowledgments:** The authors give thanks the agencies that support this research and all the support personal from Universidade Federal de Pelotas and the University of Arkansas.

**Conflicts of Interest:** The authors declare no conflict of interest.

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
