# Peer review of "Rapid Reduction of Herbicide Susceptibility in Junglerice by Recurrent Selection with Sublethal Dose of Herbicides and Heat Stress"

_agronomy, doi:10.3390/agronomy10111761_

Round 1

Reviewer 1 Report

The idea of this study is interesting and important to me. The level of treatment and the focus on interaction of two factors over generation of weeds are appropriate for weed science community. However, technical quality of this manuscript is poor. Too much problem with all sections in this manuscript (intro, M&M, results and discussion). I have marked many serious drawbacks in the reviewer PDF i attached. Intro and discussion need to be rewritten completely. Why four herbicides used in this study and why rates were selected have never been justified or explained at all. Pots staying in greenhouse for 30C but counterparts stayed in growth chamber for 45C raised concern from me too, as we can never reproduce sunlight and temperature change inside the greenhouse in a spray chamber. Data presentation can be improved to be more reader friendly. Lots of numbers as "results" could not be found in table, figure or anywhere. Figure 1 was not mentioned at all in the result section and there was not a figure 2 which was mentioned multiple time. Not sure where data derived from figure 2 came from. Moreover, if G3 showed most tolerance to herbicides among all three generations, why using G2 for analysis of gene expression??? This is not a wise move in my opinion. 

Although the study addresses an important aspect of weed science, the technical quality of the manuscript does not warranty acceptance in such format.

Author Response

Cover letter

Dear Reviewer 1,

            First of all, thank you very much for all your comments and for reviewing thoroughly the manuscript. We split the general comment to answer the questions point by point.

Reviewer 1 comments

General comment from Reviewer 1:

Comment: The idea of this study is interesting and important to me. The level of treatment and the focus on interaction of two factors over generation of weeds are appropriate for weed science community. However, technical quality of this manuscript is poor. Too much problem with all sections in this manuscript (intro, M&M, results and discussion). I have marked many serious drawbacks in the reviewer PDF i attached.

Response: We agree that the manuscript have seral point to be enhanced and we took into account all the suggestions pointed in the PDF.

Comment: Intro and discussion need to be rewritten completely. Why four herbicides used in this study and why rates were selected have never been justified or explained at all.

Response: Regarding four used herbicides, we added a justification in the text. “The treatments tested in this study were based in optimal (30 °C) and high temperature (45 °C) and four active ingredients herbicides recommended to E. colona control in rice crop (Table 1).“

Comment: Pots staying in greenhouse for 30C but counterparts stayed in growth chamber for 45C raised concern from me too, as we can never reproduce sunlight and temperature change inside the greenhouse in a spray chamber.

Response: We used a greenhouse for 30 °C experiment because we had only one growth chamber available and the greenhouse presented all the controls (temperature, humidity; light, photoperiod, etc). Also, we could not perform the 45 °C in the greenhouse together with the 30° C treatment because the greenhouse was shared with other experiments needing the optimal conditions for E. colona and other plants growing.

Comment: Data presentation can be improved to be more reader friendly. Lots of numbers as "results" could not be found in table, figure or anywhere. Figure 1 was not mentioned at all in the result section and there was not a figure 2 which was mentioned multiple time. Not sure where data derived from figure 2 came from. Moreover, if G3 showed most tolerance to herbicides among all three generations, why using G2 for analysis of gene expression??? This is not a wise move in my opinion.

Response: We improved the data presentation. Inserting the gene expression values and the confidence interval inside the heat map figure. The figure 1 is a scheme of the experiment and was cited in less frequency compared to figure 2 which demonstrates the dose-response curve for all tested herbicides. The data of figure two are described and can be verified in the table 3, where you can find the estimated parameters fitted with a three-parameter log-logistic regression model in equation 1 and figure 2. We do not generate a G3 population, we generated G0, G1 and G2 population being G2 less susceptible and was used for gene expression analysis.

Comment: Although the study addresses an important aspect of weed science, the technical quality of the manuscript does not warranty acceptance in such format.

Response: We improved the manuscript to address all the questions and concerns raised by the reviewer and hopefully it can be acceptable in the current format.

Comment: line 43-47: This statement has little relevance to the study and junglerice, it should be deleted.

Response: We think that this statement is suitable and it was a consensus among all the authors.

Comment:  line 48: for example????

Response: We clarify in the text

Comment: line 52-57: These highlighted statement has little relavance to E. colona (junglerice) of this study. They are too broad, do not provide much new or useful information and should be deleted as well. Wordy introduction that do not hit points can discourage readers continue reading this paper.

Response: We think that this statement is suitable and it was a consensus among all the authors.

Comment: line 62-63: Wordy statement. Consider revise it to "The first case of herbicide-resistant weed of any species in this genus was E. colona, reported in 1987 in Costa Rica, to propanil, a photosystem II inhibitor."

Response: We revise this statement.

Comment: line 65: to change: “are reported” to have been reported

Response: We change accordantly.

Comment: line 67: “weed groups”. What is the definition of weed group? Do you mean genus? Weed group is not a common term.

Response: Sorry, we change for “weeds” only.

Comment: line 72: “unintentionally”. Low dose selection can totally be occur intentionally. I can name a dozen growers i worked with who have applied reduced rate in the past. Saving herbicide cost is a main driver for reduce rate application.

Response: We remove the word “unintentionally”, because you are right. The 'Low-dose selection’ of weed populations occurs in the field all the time and can occur both, unintentionally and with intention.

Comment: line 77: “Maximum herbicide efficacy is highly dependent on optimum environmental conditions [22]”. This sentence here does not flow at all with the rest of paragraph which is about low dose selection.

Response: We change this sentence.

Comment: line 78:The increase in global average temperature”. I don't disagree global climate change can increase average temperature. But I can also say for sure that hot dry summer makes killing annual grasses like junglerice way more difficult. Hot, dry and no rain conditions happen all the time. Hot and dry conditions provide another level of justification for this study.

Response: We rewrote this sentence to fit better with the justification of our study.

Comment: line 86: What does NTSR indicate?

Response: We inserted in the text what NTSR indicate.

Comment: line 95: And why four postemergence herbicides were chosen in this study, not the others? Does any E. colona population evolved resistance or enhanced tolerance to these herbicides already?

Response: We chose four active ingredients herbicides recommended to E. colona in rice crop, one is new synthetic auxins (florpyrauxifen-benzyl), the other is an old active ingredient (quinclorac) and cases of resistance have been reported in E. colona.

Comment: line 95: Why focus on this weed species? What is the significance of this weed species if they are better managed in field?

Response: We focus in E. colona specie because is one of the most important weed present in many crops and world regions causing significant losses in yield, high competitive ability, fast growing and have been reported globally to herbicide resistance.

Comment:  line 95: Why focus on certain genes but not the others? What has been done on these genes which may confer increased herbicide tolerance? 

Response: We chose genes reported to be involved in herbicide metabolism and abiotic stress tolerance.

Comment:  line 96: “with transgenerational effect“. Awkward sentence with too much fragments. Also, what is the exact definition of transgenerational effect? It was not clearly defined or explained in the intro and no related literature was covered either. Can weed tolerance to herbicide (not target site mutation) be passed down to future generations? Authors should review literature to better explain what we know about transgenerational effect, instead of just throwing out a term in hypothesis.

Response: transgenerational effect: be pass down to future generations.

Comment:  line 105: “using seeds of susceptible E. colona”. How they were tested for susceptibility? Susceptible to what herbicide at what rate?

Response: seeds were tested for all used herbicides at recommended dose.

Comment: line 106: “from a field”. What field this was? Crop field, pasture, hay field, vegetable farm, etc. Have the four herbicides used in this study been applied here in the past?

Response: seeds were from rice filed without herbicide application historic.

Comment:  line 106-109: This paragraph should only be used to describe how plant material/junglerice population was collected. This highlighted statement seems to be redundant to me, considering detail procedures are described below.

Response: We divided it into topics to facilitate the reading for general audience, as observed in many other articles in the field.

Comment: line 112: “Seeds”. How many seeds? How many per cell?

Response: We added this information in the text.

Comment: line 114: “seedlings were transplanted into square pots”. How many seedlings per pot?

Response:  We added this information in the text.

Comment: line 117: “in each cycle”. Was this experiment repeated over time??? Six reps in one time will not be enough for a publication unfortunately.

Response: We used three cycles of six replicates in each.

Comment: line 121: Figure 1. There is no information about flooded and normal temperature. Why each plot needs to be flooded? How each plot was flooded? What was "normal temperature"? If it was 30C, then just use 30C instead of throwing another term here.

Response: Each treatment was flooded with a water to complete the cycle. We change normal temperature to optimal temperature.

Comment: line 124: “The numbers 30 and 45 are the temperature (°C) treatments”.  This has been clearly shown in figure 1 and no need to be mentioned again.

Response: We removed it from the figure caption.

Comment: line 127: “recommended dose”. Recommended dose for which crop? I am sure the rate is different across crops.

Response: It is a recommended dose for rice crop;

Comment: line 131: This statement should be moved to line 112. Also, when was the experiment conducted??? 

Response: We change to the begging of the paragraph.

Comment: line 134-135: There is a potential problem here. If 30C treatment stayed in greenhouse and 45C treatment was in growth chamber, there is no way the environment in greenhouse matched the environment in growth chamber 100%. You can not simulate relative humidity and sunlight change in a growth chamber as well as in a greenhouse. In my opinion, 30C treated pots should be placed in growth chamber as well to reduce environmental variances. There is a major flaw of the study.

Response: We used a greenhouse for 30 °C experiment because we had only one growth chamber available and the greenhouse presented all the controls (temperature, humidity; light, photoperiod, etc). Also, we could not perform the 45 °C in the greenhouse together with the 30° C treatment because the greenhouse was shared with other experiments needing the optimal conditions for E. colona and other plants growing.

Comment: line 137: When these plants were under heat stress, how pots were watered during this 7 day period? If they are not watered and remained at 45C for 7 days, they will be oven dried. Also, is there a temperature change in the growth chamber to simulate diurnal effect? Was 45C constant during those 7 days?

Response: Plants for both temperature treatments were irrigated daily. A temperature gradient was performed where the 45 °C was the higher temperature at midday and the temperature decrease for the next hours until night temperature.

Comment: line 141: with what type of nozzle? Spray speed used?

Response: We added this information.

Comment: line 159: “This procedure was repeated over three cycles”. I assume this statement means this experiment was repeated three times??? What does cycle means here exactly?

Response: Yes, we performed three cycles. Cycles means that we sow the seeds and we treated the seedlings and after we maintained the plants at optimal conditions until get seeds again. Each cycle was named as G0, G1 and G2.

Comment: line 167: “0.0625, 0.125, 0.25, 0.5, 0.75, 1.0, 1.5, 2.0 and 4.0x the recommended dose”. All these doses need to be specified in g ai/ha.

Response: We have four active ingredients and for this reason we do not specified in g ai/ha.

Comment: line 170: “3 wk”. use three weeks to be consistent with line 143

Response: We change for three weeks.

Comment: line 174: “Y = d/1 + exp[b(log x − log e)]”. This equation needs to stay in the middle of this line

Response: We changed to the middle of the line.

Comment: line 175: “expressed as percentage of the nontreated check”. There is no need to convert it to % of NTC because it is a percentage already. You NTC should have 0% injury so you can not convert it to % of NTC anyway. Delete this statement.

Response: We deleted.

Comment: line 189: “herbicide application (T0) and 12 h after herbicide application (T1)”. This was collected on all three generations???

Response: No, only in G0 under 30 °C and G2 in all treatments.

Comment: line 231: “a sublethal dose”. subleathal doses. They were treated with more than one subleathal dose.

Response: We change.

Comment: line 236: “ED50 increased 0.12 points”. How can ED50 increase by points? ED50 has unit which is g ae ha-1. What does point mean here? Fix all the similar issue in the next paragraph.

Response: We deleted “points”.

Comment: line 237: “Figure 2A-C”. There is not a figure 2 in this manuscript. Not sure what authors refers to here.

Response: We think a problem with the file occur. Figure 2 is related to dose-response curve.

Comment: line 239: “the resistance factor remained unchanged“. There is no information regarding resistance factor in table 3. Please elaborate this statement.

Response: Yes, it is about the tolerance index.

Comment: line 241-242: “orders of magnitude higher“. orders of magnitude higher? Highest ED50 of G2 under 45C was about twice higher than ED50 of G0. I have a hard time to qualify two times increase as "orders of magnitude higher". It wasn't 10 times higher or anything like that.

Response: We change to “increase were higher compared to…” only.

Comment: line 242-245: I can not find figure 2 in the manuscript so i am not sure what authors are saying here about increase by xx points.

Response: We think a problem with the file occur. Figure 2 is related to dose-response curve.

Comment: line 245-246: “10 to 37”. How did authors come up with these numbers??? ED50 increased about two times at the most in terms of g ae ha-1 in table 3. Where 37 time increase came from???

Response: Glufosinate G0 30 °C ED50 53.6/5.28 (increase)=10 times; Quinclorac 18.4 increase/0.5=37.

Comment: line 247: What about figure 1? It is not even mentioned in the manuscript.

Response: Figure 1 in respect to the experiment scheme, it is cited in the methods section.

Comment: line 247: There is no discussion at all here about how data in table 3 correlate to any published study findings. Typical problem in many graduate student's papers including mine.

Response: discussion of table 3 and figure 2 is the same.

Comment: line 264: change name Figure 1 to Figure 2.

Response: Figure 1 appears only during the methods section.

Comment: line 280: “0.7-fold”. 0.7 fold can be confusing. Some readers without molecular background may consider that is 30% reduction compared to control plants. 0.7 Fold increased expression means 70% increase compared to control? A better explanation is needed.

Comment: line 293: “-0.1-fold”. -0.1 fold means 10% reduction or what?

Comment: line 313: “-1.4”. authors need to explain negative fold number like this one here better, at least in table footnote.

Response: The data is converted in log2, in this sense it is assumed that the control present 0.0 expression while 0.7 means 0.7 higher than 0.0, to be 70% higher we need to find an expression of 70-fold. Fold change is a measure describing how much a quantity changes between an original and a subsequent measurement. It is defined as the ratio between the two quantities. Any value higher than 0.0 means increasing in expression while negative numbers or lower than 0.0 means decreasing or repression of an expression.

Comment: line 313: change word form to from. this should be from, not form

Response: We changed.

Comment: line 316: “Figure 3. Legend…” What does -4 to 9 indicate in this scale? Is the value a ratio or it actually has a unit of measurement? Need to explain this better in footnote.

Response: -4.0 was the lower expression value found among the tested genes while 9.0 means the higher value in expression found among the tested genes,

Comment: line 323-24: “T0 = before herbicide application; T1 = 12 h after herbicide application to G2 plants”. before herbicide application on G0 generation? What is the baseline to compare to in this figure? Before application on G0 or G2 plants???

Response: mRNA abundance of each gene from G0 population (control from susceptible standard) served as the baseline for determining relative RNA levels, in which treatment from G2 population.

Comment: line 323-24: Why not using G3 plants? They have shown even better tolerance and high ED50 in table 3.

Response: We do not generate G3 plants, we got only G2 plants.

Comment: line 384: “candidate genes”. These candidate gene expression was increased following heat stress and herbicide? If so, state clearly here.  Has anyone found similar results in their study???

Response: These candidate genes were previously reported to be expressed in response to herbicide and/or heat stress. Yes, some previous reports found similar results which are described in the discussion section.

Comment: line 386: Authors should use a table to clearly show which gene has increased or decreased expression over generations, and at what level. I am not a big fan of all the "fold" here. The baseline that these fold compared to was not clear to me either. What about just using % increase or decrease in this table?

Response: Express gene expression in fold is a common way to show gene regulation in molecular papers, we inserted inside the figure 3 the log2-fold change numbers in each treatment.

Comment: line 398: “the recommended dose”. recommended dose in this paper were never very clear. Where they came from? for which crop?

Response: from rice crop for E. colona control found in herbicide label.

Comment: line 400: “as dramatic as”. How dramatic? Use numbers to illustrate the level reported in these literature.

Response: We changed to “higher than”

Comment: line 409-10: “favor the selection of Echinochloa spp. resistant plants [15,25]”. favors for more resistant plants to develop, or favors more herbicide tolerance as found in this study? Target site mutation is different term compared to enhanced expression of certain enzymes.

Response: We changed this sentence in the text.

Reviewer 2 Report

Major Comments:

Overall this seems like a well conducted study. The methods are generally appropriate, although more details are needed in the Methods section, and the conclusions are in line with the results.

While the writing is generally clear, the manuscript needs to be thoroughly edited by a native English speaker. Word choices are sometimes odd (the use of “obtained”, for instance), and the grammar is often awkward.

Minor Comments:

line 117: The experiment did not follow a completely randomized design, at least regarding temperature, because plants in the 45C treatment were all in the same incubator.

lines 126-127: More explanation is needed for why these herbicides were chosen. Because they are the most commonly used against this weed, I'm guessing?

lines 134-137: Only one growth chamber was used for the 45C treatment, correct? Why wasn’t a growth chamber also used for the 30C treatment? Were conditions besides temperature, such as humidity, similar between the growth chamber and the greenhouse?

lines 138-145: This passage about the preliminary trial would fit better at the start of the paragraph.

lines 166-167: How many plants from each generation and treatment were subjected to each application rate? The caption for Figure 1 suggests that two plants were, but this needs to be explicitly stated here.

line 180: “Without stress” is too vague. I’m assuming you mean plants that were not subjected to herbicide, but do you mean plants from 30C only, or plants from both 30 and 45C?

lines 257-258: If P-values are for comparisons between temperatures, shouldn’t there be only one value in the Table for each 30/45C pair? If possible, it would be best to report actual values, rather than just “<0.05”.

Author Response

Cover letter

Dear Reviewer 2,

            Your comments were important to improve and ensure the quality of the manuscript. Below we list the reviewer’s comments and our responses.

Reviewer 2 comments

Reviewer comment: Overall this seems like a well conducted study. The methods are generally appropriate, although more details are needed in the Methods section, and the conclusions are in line with the results. While the writing is generally clear, the manuscript needs to be thoroughly edited by a native English speaker. Word choices are sometimes odd (the use of “obtained”, for instance), and the grammar is often awkward.

Authors: We added more details in methods section. We replaced the word “obtained” for a most suitable one.

Reviewer comment: line 117: The experiment did not follow a completely randomized design, at least regarding temperature, because plants in the 45C treatment were all in the same incubator.

Authors: Sorry, we made a mistake in the sentence, the experiment was performed in complete randomized block design.

Reviewer comment: lines 126-127: More explanation is needed for why these herbicides were chosen. Because they are the most commonly used against this weed, I'm guessing?

Authors: Yes, we added the explanation in the text “General procedure for population generation”.

Reviewer comment: lines 134-137: Only one growth chamber was used for the 45C treatment, correct? Why wasn’t a growth chamber also used for the 30C treatment? Were conditions besides temperature, such as humidity, similar between the growth chamber and the greenhouse?

Authors: We used a greenhouse for 30 °C experiment because we had only one growth chamber available and the greenhouse presented all the controls (temperature, humidity; light, photoperiod, etc). Also, we could not perform the 45 °C in the greenhouse together with the 30° C treatment because the greenhouse was shared with other experiments needing the optimal conditions for E. colona and other plants growing.

Reviewer comment: lines 138-145: This passage about the preliminary trial would fit better at the start of the paragraph.

Authors: We changed the sentence related to the preliminary trial at the start of the paragraph.

Reviewer comment: lines 166-167: How many plants from each generation and treatment were subjected to each application rate? The caption for Figure 1 suggests that two plants were, but this needs to be explicitly stated here.

Authors: Yes, we added this information in the text.

Reviewer comment: line 180: “Without stress” is too vague. I’m assuming you mean plants that were not subjected to herbicide, but do you mean plants from 30C only, or plants from both 30 and 45C?

Authors: We changed this sentence in the text to “ED50 of the G0 at 30 °C”.

Reviewer comment: lines 257-258: If P-values are for comparisons between temperatures, shouldn’t there be only one value in the Table for each 30/45C pair? If possible, it would be best to report actual values, rather than just “<0.05”.

Authors: Yes, we changed the p-values keeping only one in each 30/45 °C pair and also added the actual values.

Reviewer 3 Report

The manuscript entitled ‘Rapid reduction of herbicide susceptibility in junglerice by recurrent selection 2 with sublethal dose of herbicides and heat stress’ concerns the combined effect of heat stress (HS) and sublethal dose of herbicides (four active ingredients) on adaptive gene expression and herbicide efficacy on Echinochloa colona (junglerice);

The subject of the manuscript falls with the general scope of the Journal and provides interesting data for the scientific community. In general it was well conducted, but in the perspective of an environmentally friendly management of agro-ecosystems I suggest to add some information about the use of bioherbicide just for Echinochloa colona, whereas it is one of the most troublesome and almost ubiquitous weed species in rice crops. Moreover, bioherbicides can be used in combination with synthetic herbicides, with the aim of reducing the dosage. Therefore I suggest adding this speculation in the Introduction and/or Discussion Sections.

KEYWORDS: In order to improve the attractiveness of the manuscript, I suggest to add ‘herbicide’

ABSTRACT is concise; INTRODUCTION: the background and problem are clearly stated but I suggest to add a brief mention of the possible use of bioherbicides; MATERIALS AND METHODS and RESULTS are generally acceptable, but lacks of detailed statistical information; CONCLUSIONS are in line by the obtained results; REFERENCES in my opinion there too many references (103 references), I suggest reducing the number. Tables and Figures are appropriate.

STATISTICAL ANALYSIS: In addition of figure captions, please add detailed information about the statistical procedure in the M&M. In particular, specify the type of regression used and validation of the basic assumptions (homoscedasticity, normality and trasformations of data if any).

I have made a few suggestions in "Minor Considerations". The following point may be considered while revising the manuscript:

- In the Introduction Section, I suggest to add a brief mention for the use of bioherbicides in crops. To improve the bibliography, I strongly suggest adding this recent paper, as an example of use of natural substances for weed control:

Allelopathic potential of leaf aqueous extracts from Cynara cardunculus L. On the seedling growth of two cosmopolitan weed species (https://doi.org/10.4081/ija.2019.1373), in which are reported the results for sustainable alternatives to synthetic herbicides in weed control. The utilisation of plant extracts as possible bioherbicides represents an important solution to avoid the accumulation of dangerous compounds in the soil, especially for food security.

Furthermore, the combined use of bioherbicides and herbicides within an integrated weed management approach is widely reported in literature. In this regard, I may suggest the recent review:

Integrated Weed Management in Herbaceous Field Crops (https://doi.org/10.3390/agronomy10040466)

Author Response

Cover letter

Dear Reviewer 3,

            Your comments were important to improve and ensure the quality of the manuscript. Below we list the reviewer’s comments and our responses.

Reviewer 3 comments

Reviewer 3: KEYWORDS: In order to improve the attractiveness of the manuscript, I suggest to add ‘herbicide’

Authors: We add ‘herbicide’. Line 35

Reviewer 3: ABSTRACT is concise; INTRODUCTION: the background and problem are clearly stated but I suggest to add a brief mention of the possible use of bioherbicides; MATERIALS AND METHODS and RESULTS are generally acceptable, but lacks of detailed statistical information; CONCLUSIONS are in line by the obtained results; REFERENCES in my opinion there too many references (103 references), I suggest reducing the number. Tables and Figures are appropriate.

Authors: We insert more details regarding the statistical analysis. Related to reference number, the journal does not present a maximum number of citations. In this sense, we demonstrated the state of the art and those important articles with fit our findings to bring a deep explanation and overview of our findings.

Reviewer 3: STATISTICAL ANALYSIS: In addition of figure captions, please add detailed information about the statistical procedure in the M&M. In particular, specify the type of regression used and validation of the basic assumptions (homoscedasticity, normality and trasformations of data if any).

Authors: We added a sentence related the basic assumptions and the type of the regression was stated in the text and in the table 3 and figure 2: “three-parameter log-logistic model”.

Reviewer 3: I have made a few suggestions in "Minor Considerations". The following point may be considered while revising the manuscript:

- In the Introduction Section, I suggest to add a brief mention for the use of bioherbicides in crops. To improve the bibliography, I strongly suggest adding this recent paper, as an example of use of natural substances for weed control:

Allelopathic potential of leaf aqueous extracts from Cynara cardunculus L. On the seedling growth of two cosmopolitan weed species (https://doi.org/10.4081/ija.2019.1373

), in which are reported the results for sustainable alternatives to synthetic herbicides in weed control. The utilization of plant extracts as possible bioherbicides represents an important solution to avoid the accumulation of dangerous compounds in the soil, especially for food security.

Furthermore, the combined use of bioherbicides and herbicides within an integrated weed management approach is widely reported in literature. In this regard, I may suggest the recent review:

Integrated Weed Management in Herbaceous Field Crops (https://doi.org/10.3390/agronomy10040466)

Reviewer 3: The subject of the manuscript falls with the general scope of the Journal and provides interesting data for the scientific community. In general it was well conducted, but in the perspective of an environmentally friendly management of agro-ecosystems I suggest to add some information about the use of bioherbicide just for Echinochloa colona, whereas it is one of the most troublesome and almost ubiquitous weed species in rice crops. Moreover, bioherbicides can be used in combination with synthetic herbicides, with the aim of reducing the dosage. Therefore I suggest adding this speculation in the Introduction and/or Discussion Sections.

Authors: We search in many research and review papers related to the studied topic and we do not found a way to insert this theme mainly with respect to our objectives. Our research has as main focus the recurrent selection of herbicides and abiotic stresses in resistance evolution in E. colona. The use of bioherbicides is very interesting and together with the obtained results showed in this paper can be further studied as respect to resistance evolution in E. colona and also other weeds and crop breeding.
